# Ring synthetic chromosome V SCRaMbLE

Juan Wang[1,2], Ze-Xiong Xie [1,2], Yuan Ma[1,2], Xiang-Rong Chen[1,2], Yao-Qing Huang[1,2], Bo He[1,2], Bin Jia [1,2], Bing-Zhi Li[1,2] & Ying-Jin Yuan [1,2]

Structural variations (SVs) exert important functional impacts on biological phenotypic diversity. Here we show a ring synthetic yeast chromosome V (ring_synV) can be used to continuously generate complex genomic variations and improve the production of pro-deoxyviolacein (PDV) by applying Synthetic Chromosome Recombination and Modification by LoxP-mediated Evolution (SCRaMbLE) in haploid yeast cells. The SCRaMbLE of ring_synV generates aneuploid yeast strains with increased PDV productivity, and we identify aneuploid chromosome I, III, VI, XII, XIII, and ring_synV. The neochromosome of SCRaMbLEd ring_synV generated more unbalanced forms of variations, including duplication, insertions, and balanced forms of translocations and inversions than its linear form. Furthermore, of the 29 novel SVs detected, 11 prompted the PDV biosynthesis; and the deletion of uncharacterized gene *YER182W* is related to the improvement of the PDV. Overall, the SCRaMbLEing ring_synV embraces the evolution of the genome by modifying the chromosome number, structure, and organization, identifying targets for phenotypic comprehension.

---

[1] Key Laboratory of Systems Bioengineering (Ministry of Education), School of Chemical Engineering and Technology, Tianjin University, 300072 Tianjin, People's Republic of China. [2] SynBio Research Platform, Collaborative Innovation Center of Chemical Science and Engineering (Tianjin), Tianjin University, 300072 Tianjin, People's Republic of China. These authors contributed equally: Juan Wang, Ze-Xiong Xie. Correspondence and requests for materials should be addressed to Y.-J.Y. (email: yjyuan@tju.edu.cn)

Genomic SV refers to heritable nucleotide sequence differences of at least 50 base pairs (bp) in size[1], and has long been interpreted with respect to phenotypic diversity and human disease[2–8]. Chromotropic rearrangements induced by a neochromosome especially arise in a ring chromosome structure[9–11]. Thus, genomic SVs not only exert their functional impact on phenotypes, but are also implicated in genotypic evolution with spatial interactions. Strategies have been adopted to enable genetic changes of SVs to provide a highly tractable approach for the functional analysis of genomic rearrangements. Especially, clustered regularly interspaced short palindromic repeats (CRISPR)- CRISPR-associated protein 9 (Cas9) technology has been applied to successfully generate chromosomal rearrangements implicated with cancer, as well as recover large chromosomal inversion for functional correction[12,13]. The existing genomic engineering strategies accelerated the gene editing at specific targets accurately, but was not that efficient to analyze combinational rearrangement of large DNA segments in genome scale. Thus, a rapid, simple, and highly efficient method for large DNA modification in genome level is critically needed to generate complex genome rearrangement.

The synthetic yeast chromosome has provided us the capabilities to rebuild the genome organization by the SCRaMbLE system[14,15]. The genome-wide inserted nondirectional loxPsym sites, 3 bp after the stop codon of nonessential genes and at major landmarks[16], make it possible to facilitate inducible recombination events in either direction that lead to site-specific rearrangements between loxPsym sites. The SCRaMbLE system remains dormant until the induction of the Cre recombinase, which should in principle generate desirable genomic variations including deletions, inversions, translocations, and duplications at multiplex genetic locations simultaneously[17]. Indeed, deletion, inversion, insertion, and duplication were all detected in SCRaMbLEd linear chromosomes both in haploids and diploids[16,18,19], though limited rearrangement events occurred and the majority of the chromosome remained unchanged. Relatively complex rearrangement events were observed when SCRaMbLEing the synthetic yeast chromosome arm synIXR, which is incorporated in a bacterial artificial chromosome (BAC)[17]. Therefore, we presume that the chromosome topological structure exerts its functional impact on chromosome rearrangement and the linear chromosome is not an ideal platform to generate combinational genetic backgrounds.

Here, to continuously evolve strains of defined phenotype and genotypic SVs, we perform the inducible chromosome rearrangement by SCRaMbLEing yeast haploid cells carrying ring_synV. The only difference between the ring_synV and synV is the telomere removal[20]. The 534,637-bp ring_synV carrying strain displayed no notable growth defects and is expected to be Cre-hypersensitive since the whole chromosome was divided into 170 segments by the 170 integrated loxPsym sites. The PDV is not only a precursor of potential anti-cancer drug violacein with a color of dark green for visual screen, but also a downstream metabolite of aromatic amino acids (AAA), which are important precursors of many high value-added biochemicals, drugs, and nutraceuticals. We use the PDV biosynthesis pathway as a selective marker to define the genotypic diversity by combinatorial rearrangements. The ring_synV strain represents an ideal genetic background to generate inducible chromosome rearrangements while the following deletion of essential genes or occurrence of synthetic lethal genetic interactions will cause cell death; the surviving colonies could refract the fluctuation of PDV production. We isolate derivatives with a broad variety of phenotypes, facilitated by structural variants especially the large DNA segment duplication, translocation, and inversion. We specially identify and investigate the SVs may correlated with PDV metabolism improvement, identifying targets for phenotypic comprehension.

Additionally, we demonstrate that chromosome with topological changes have the capability for further SV, and the cells with a moderate intensity of phenotype reinforcement are potential for continuous phenotypic evolution. The SCRaMbLE method is generally useful for alteration of defined phenotype and genotype with a ring chromosome genetic background, especially for genome reorganization.

## Results

**Continuous phenotypic evolution by SCRaMbLEing ring_synV.** The ring_synV chromosome is divided into 170 segments by 170 loxPsym sites and is capable to generate massive combination of genetic diversity when induced by Cre recombinase. We integrated a PDV biosynthesis pathway at the *CAN1* locus (segment 7) to define the productive diversity of PDV and the genetic rearrangement of ring_synV in haploid yeast cells by color changes (Fig. 1 and Supplementary Data 1). During the induction, SCRaMbLE events frequently occurred and exhibited an observed high death rate, since deletion of essential genes or the occurrence of synthetic lethal genetic interactions will cause cell death[21]. Post-SCRaMbLE, the surviving colonies with different colors could refract the fluctuation of PDV production; the color changes of surviving colonies represent the rearrangement of ring_synV. Since the synV has been demonstrated with a perfect potential to generate diverse phenotypes by SCRaMbLE, we first compared the SCRaMbLE result of ring_synV with synV by using pCre4 plasmid[19,22].

Haploid cells carrying a ring_synV performed a similar tendency of phenotypic improvement, with the synV when exposure to estradiol. We assigned the color of initial strain (yWJS1) with the score of 0 and all the surviving SCRaMbLEd strains were evaluated as darker, original, lighter and white, representing the increased, unchanged, reduced, and void of PDV production, respectively. Upon estradiol addition, we observed that the maximum colony ratio with a relative darker color to the parent strain appeared when cells were exposed to estradiol for 2 h (Supplementary Fig. 1a–c). We counted the number of surviving colonies with darker colors relative to the whole surviving colonies, reporting a darker frequency. Nearly 3.67% of ring_synV and 2.61% of synV survivals seems to have an optimized PDV production, respectively. It is possible that the chromosome rearrangement occurred only once during the cell lifespan. However, the ratio decreased to 1.04% and 0.86% when exposed to estradiol for 4 h. Following, another peak of darker colonies was achieved as the exposure time increases to 8 h, revealing that the percent of colonies with improved phenotypes being increased by the influence of chromosome rearrangement. Meanwhile, we also observed that the minimum percent of colonies with a relative lighter color was achieved at 4 h. Especially, for the colonies with a lighter color, the synV exhibited a relatively dramatic increase when induced for 8, 12, and 16 h, and then showed no significant increase. Interestingly, a higher ratio of lighter colonies was generated in the ring_synV background when exposed to estradiol for 24 h, which manifested that the ring_synV performed with greater variability and may tolerate more genetic rearrangement. We also observed a lower percentage of lighter colonies in haploid cells carrying a ring_synV when induced for 8 and 12 h. Especially, when exposed to estradiol for 24 h, more colonies with lighter colors were observed in the ring_synV background, while no significant increase in the strains carrying synV (Supplementary Fig. 1b, c and Supplementary Data 2).

Four of the surviving colonies with optimized PDV production were selected for the second round of SCRaMbLE, yWJS007, yWJS039, yWJS047, and yWJS067. We counted the number of surviving colonies and evaluated their phenotypes by comparing

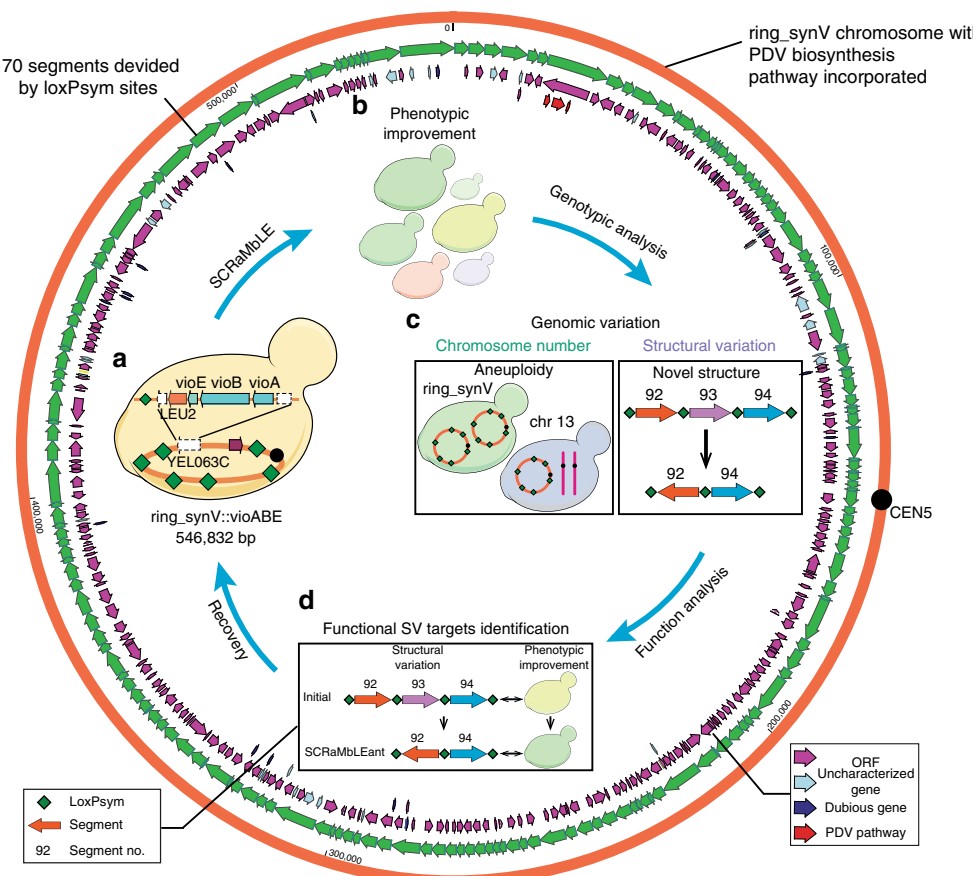

**Fig. 1** Ring_synV SCRaMbLE to generate genomic and phenotypic variation. The ring_synV (orange circle) was divided into 170 segments (green arrows) by loxPsym sites, in which genes were demonstrated in purple and blue arrows. **a** The PDV biosynthesis pathway was incorporated in *YEL063C* (*CAN1*) of ring_synV by homologous recombination to generate the initial PDV producer strain yWJS1. **b** The SCRaMbLE system of yWJS1 functioned and SCRaMbLEd colonies with diversity of PDV improvement were isolated when exposed to galactose and estradiol. **c** Genomic SV analysis of SCRaMbLEants revealed chromosome number variation, CNV and novel SV. **d** Colonies with increased PDV production were selected when transformed novel SVs into the initial strain

with their initial strains (Supplementary Fig. 2a, b). Although a majority of the surviving colonies retained unchanged color, upon visual inspection, we observed 2.81%, 3.54%, 21.07%, and 40.92% colonies with darker color for the four candidates, respectively. For yWJS007, yWJS039, and yWJS067, more than 72% of the surviving colonies retained unchanged color, and only about 40.84% of surviving colonies were color unchanged for yWJS047. We observed that ~21.07% of surviving colonies were darker for yWJS039 and ~40.92% for yWJS047, whereas only a few percent of darker colonies were recovered from yWJS007 and yWJS067. Notably, the percentage of colonies with darker colors in yWJS047 is ~40.92%, which is about two times to yWJS039 (~21.07%) and 15 times to yWJS067 (~2.8%) (Supplementary Fig. 2a). Though some SCRaMbLEd colonies appeared a darker color in visual, the production of PDV may be not increased. We picked 24 SCRaMbLEants with darker color from yWJS007, yWJS039, yWJS047, and yWJS067 (6 SCRaMbLEants from each parent strain) for PDV quantitatively measurement by high-performance liquid chromatography (HPLC), and we recovered an isolate with the most PDV productivity (yWJS168) from yWJS067, which was used for further SCRaMbLE.

Ring_synV exhibited continuous phenotypic improvement by SCRaMbLEing (Fig. 2a, b). We carried out another 5 rounds of SCRaMbLE experiment with a SCRaMbLEed strain (yWJS067), expecting that viable colonies should also yield rearranged chromosomes and the production of PDV should raise. For the

final selected SCRaMbLEed strain, the percentage of darker colonies were 7.79%, 3.76%, 4.09%, 3.88%, 3.84%, and 1.07% (Supplementary Fig. 2c, d). In each round of SCRaMbLE, though some colonies exhibited a darker color, the real productivity of PDV were not increased. For each round of SCRaMbLE, strains that had the visual darkest color and similar growth fitness comparing with the parent strain were selected for quantitation of PDV productivity by HPLC. Interestingly, parts of the selected strains exhibited a decrease of PDV production though they have visual darker colors. After six cycles of SCRaMbLE exhibition, we obtained an isolate with more than sevenfold increase in PDV production as well as a near-wild-type growth (Fig. 2a). Although some colonies exhibited a very darker color, they were not selected for further genetic analysis or high-titer production of PDV due to growth defects. Therefore, colonies with wild-type growth were selected for further genomic analysis of undergone diverse chromosome rearrangement (Fig. 2a, b and Supplementary Data 3).

**Continuous genotypic evolution by SCRaMbLEing ring_synV.** To deeply reveal the genetic variations generated in ring_synV, we selected the set of SCRaMbLEants for whole-genome sequencing (WGS) analysis, yWJS044, yWJS047, yWJS067, yWJS184, and yWJS321. Among the analyzed SCRaMbLEants, yWJS044, yWJS047, and yWJS067 were generated in the first round of SCRaMbLE from the initial strain (yWJS1), yWJS184 was generated

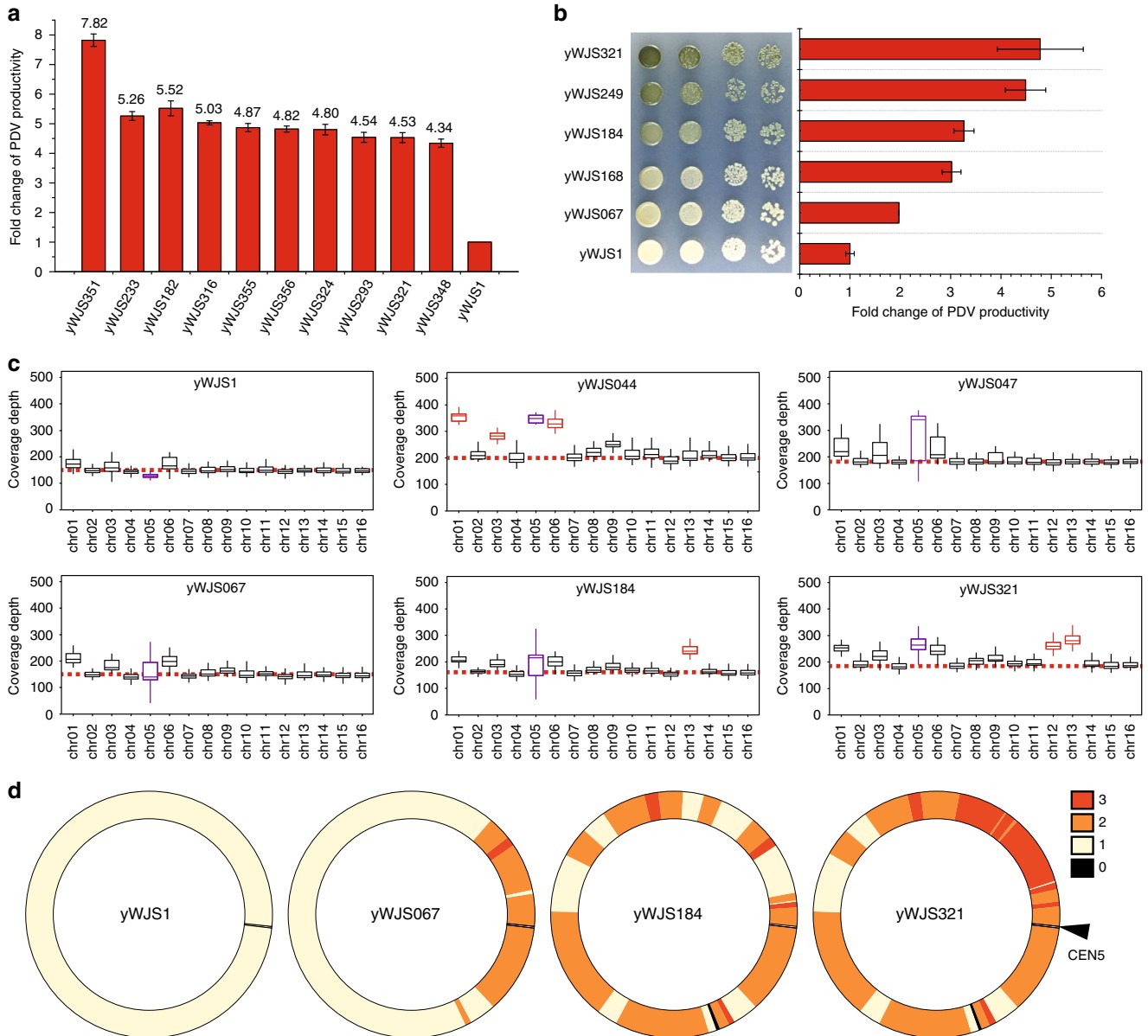

**Fig. 2** Continuous improvement of PDV production and generation of SVs. **a** Fold changes of PDV production in the top 10 SCRaMbLEd strains. **b** Phenotypic analysis of SCRaMbLEants and fold changes of PDV production after each round of SCRaMbLE. Initial strain, yWJS1; first cycle SCRaMbLEant, yWJS067; second cycle, yWJS168; third cycle, yWJS184; forth cycle, yWJS249; fifth cycle, yWJS321. **c** Boxplots showing WGS read cover depth, which indicated recurrent chromosomal aneuploidies by SCRaMbLEing the ring_synV in haploid yeast strain. The strains yWJS044 (aneuploid chromosomes: I, III, ring_synV and VI), yWJS047, and yWJS067 were generated by SCRaMbLEing initial strain yWJS1. The strains yWJS184 (aneuploid chromosome: XIII) and yWJS321 (aneuploid chromosomes: XII and XIII) were generated by SCRaMbLEing yWJS067. The coverage depth stands for the chromosome CN that were determined from WGS result. The aneuploid chromosomes were marked in red, and ring_synV was marked in purple. The centre line in each box denotes the median of read coverage depth, the upper bound of box denotes the upper quartile and the lower bound denotes the lower quartile, the upper whisker denotes the maximum read coverage depth and the lower whisker denotes the minimum read coverage depth. **d** CNV heatmap of ring_synV. Read depth-based CN estimations (loxPsym divided segments) are indicated by color (scale provided to the right). Values are averages from three experiments, and error bars denote s.d. Strain information was listed in Supplementary Data 3

in the third round of SCRaMbLE from yWJS067, and yWJS321 was generated in the fifth round of SCRaMbLE from yWJS184.

The WGS result revealed recurrent aneuploidy of specific chromosomes in yWJS044, yWJS184, and yWJS321, presenting abnormal numbers of chromosomes in the cell (Fig. 2c). We identified four aneuploid chromosomes in yWJS044, including chromosome I, III, VI, and ring_synV. Though no aneuploid chromosome was identified in yWJS067, the chromosome XIII was aneuploidy in yWJS184; the chromosome XII and XIII were aneuploidy in yWJS321. The read depth-based chromosome

number variation demonstrated that the ring_synV is an ideal genetic background to generate aneuploidy. The majority of aneuploidies may be a detrimental consequence of drastic genomic rearrangement of ring_synV. Though aneuploidies typically reduce cell fitness[23], our frequently recurring aneuploidies performed wild-type fitness comparing with their initial strains (Fig. 2b).

The result revealed long segmental duplications in all of the three SCRaMbLEants compared with its initial parent strain (yWJS1) (Fig. 2d). The WGS read depth is used to accurately

predict the copy number (CN) within ring_synV. We identified large-scale duplications at positions 60,406–77,414, 84,661–119,431, 121,405–145,169, 145,922–211,091, and 229,060–233,607; and triplication at positions 77,415–84,660 in yWJS067. The total length increased is 159,755 bp, accounting for ~29.21% of ring_synV. We constructed ring_synV CN maps across the isolates from the first (yWJS067), third (yWJS184), fifth (yWJS321) round of SCRaMbLE and their parent strain (yWJS1) at loxPsym divided segmental resolution to assay the full extent of large-scale copy number variation (CNV) and evolution. As expected, events of segmental CN change occur during the continuous chromosome rearrangement. We identified one segmental deletion and 10 large segmental CNVs across the ring_synV. For example, the duplication of region 17–24 is tripled in yWJS321. To sum up, the increased sizes of duplicated region account for 70.05% in yWJS184 and 101.02% in yWJS321 of initial ring_synV, respectively.

**Functional targets identification**. According to the WGS result, we speculate that the unbalanced form variation (especially duplication) should contribute to the high productivity of PDV. Especially, the large-scale DNA duplications strongly influenced both types of SV and CNV. In the cases of SCRaMbLEants, it is possible that the enhanced phenotype was associated with other SVs. The extensive inversions, duplications, insertions, and translocations have interfered with gene expression, or deletion of several genetic segments gives rise to a complex phenotype not previously described. WGS revealed the PDV pathway is duplicated in some SCRaMbLEd strains. However, the production has

not increased that much when an additional copy of the PDV pathway is incorporated[24]. Thus, we reasoned the enhanced phenotype was associated with SVs.

Deletion, insertion, translocation, inversion, and duplication are the most frequent genomic structural alterations by loxPsym induction. Since the genomic rearrangement did not affect internal sequences between two loxPsym sites, we focused on the novel structural junctions involving loxPsym sites. By applying the method described elsewhere, we further analyze the detailed rearrangement events in yWJS067, yWJS184, and yWJS321[17]. Aside from the 170 original loxPsym junctions in ring_synV, we figured out 53 novel structural junctions (Supplementary Fig. 3a). There are 29 novel junctions in yWJS067 and 47 novel junctions in yWJS184 and yWJS321, which indicate more complex structures were subsequent from the neochromsome during the continuous SCRaMbLE (Supplementary Fig. 3a, b). Some junctions indicated that the adjacent segment have participated in multiple events, for example, the segment 27 were revealed in both the structure 1 and 5, which is consistent with the CNV duplication (Figs. 2d, 3a and Supplementary Fig. 3). In order to determine the detailed chromosome rearrangement events and physical sequence, we applied PacBio sequencing analysis for yWJS067. The result revealed complicated inverted duplications, tandem duplication and translocated duplications (Fig. 3b).

Novel junctions with different intrachromosomal fusion types may be generated by deletion, duplication, inversion, and translocation with both head-to-head and tail-to-tail orientation. Thus, novel structural junctions may lead to an open reading frame (ORF) with a noncognate untranslated region (UTR) or

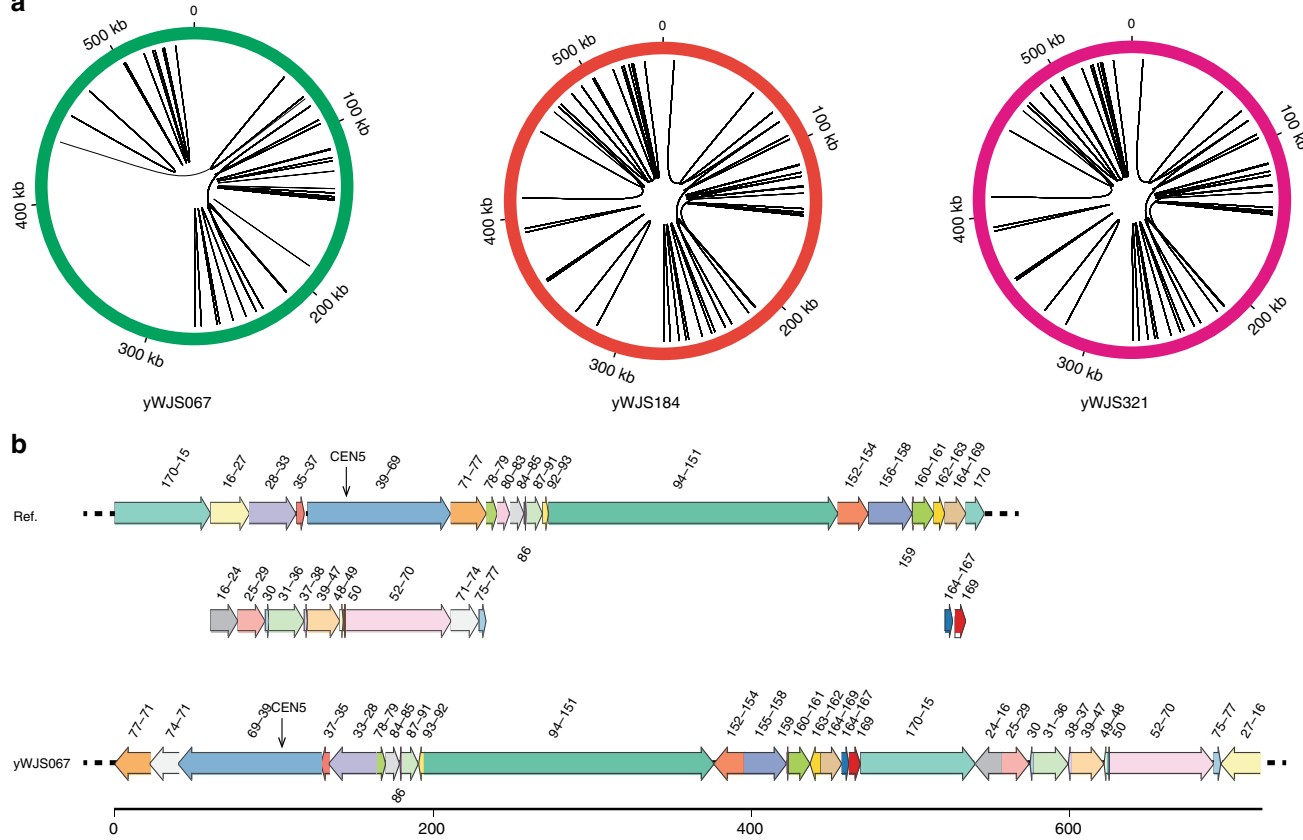

**Fig. 3** Intrachromosomal interactions in the SCRaMbLEd ring_synV. **a** CIRCOS diagrams showing the interchromosomal SV fusions for the SCRaMbLEants yWJS067, yWJS184 and yWJS321, and interacting partners with each line representing an interaction. Chromosomes are plotted across the circle. **b** The SCRaMbLEd ring_synV (yWJS067) chromosome aligned to the reference. The rearrangement events of inversion, duplication, deletion, and translocation in SCRaMbLEd ring_synV chromosome (yWJS067, bottom) are shown as diagrammatic interpretations with the reference ring_synV chromosome (top)

without terminal UTR, or may create other genomic structures with possible effects on transcription and phenotype[17]. Especially, no gene deletion was detected and the PDV pathway is not duplicated in yWJS067. To explore whether the SVs are related to the phenotypic improvement, we amplified the 29 novel junctions from yWJS067 by using polymerase chain reaction (PCR) method, and incorporating them into the pRS413 plasmid by using yeast assembly method in the initial strain yWJS1. We isolated transformants with visual darker colors by selection from 11 novel SVs than the initial strain (Fig. 4a and Supplementary Data 4). Especially, we got an isolate from the head-to-head novel structure 44 (segment 159−loxPsym−segment 160) with ~3.48-fold increase of PDV production (Fig. 4b). These results indicate that the unusual structural junctions should have a sufficient correlation with PDV production though the exact mechanism needs further investigation.

We also performed the synthetic Polymerase Chain Reaction Tag (PCRtag) analysis for deletion arraying of strains performed high PDV production with normal growth. Of the 170 segments divided by loxPsym sites, 31 contain essential genes, 14 are blank DNA segments (without genes or other genomic features in this DNA segment) and 22 contain no PCRTags. In the rest 103 segments, there's at least one nonessential gene in it and at least one pair of PCRTags designed in the gene (Supplementary Data 1). Since deletion of essential genes or occurrence of synthetic lethal genetic interactions are lethal[21]. Thus, 103 segments could be analyzed with non-redundant PCRTag array using only one PCRTag pair per ORF (Supplementary Data 1). Of 5 SCRaMbLEd clones examined with obvious phenotypic changes, 5 segmental deletions were detected by PCRTag mapping which were accurately confirmed by WGC. Although in several cases the deletion of annotated genes did not appear sufficient to cause the observed phenotype(s), the deletions of *YEL017C-A*, *YEL017W*, *YER151C*, and *YER182W* have effect on the increase of PDV production (Supplementary Fig. 4a, b).

We further investigated the stability of the SCRaMbLEants, which carrying a neochromosome V and other aneuploid chromosome(s). Long-term fitness was tested by serially culturing in synthetic complete medium lacking histidine (SC−His) medium for ~60 generations, and the persistence of PDV production was demonstrated by phenotypic analyses (Supplementary Fig. 5).

## Discussion

Classical genomic alteration approaches, such as random mutation and genomic shuffling, rely on the variation in individual genetic targets or the shuffling of homologous sequences, which is difficult to generate gene deletion, duplication, translocation, inversion, and complex genomic rearrangement events throughout the whole genome. Random mutations are efficient in generating single nucleotide variation (SNV) or short insertion-deletion (InDel) in specific genetic target, but these strategies are not capable to generate genomic rearrangements and SVs and no such attempts have been made to alter genomes at multiplex targets[25–27]. Genome shuffling is based on the protoplast fusion and have optimized usefully phenotypes by generating rearrangements mainly between allelic DNA sequences. Although it is capable to converge dominant monomer genotypes in a specific strain, genome shuffling is not able to be used for continuous evolution of gene networks or genomes[28].

The conditional inducible evolution system, SCRaMbLE, is capable to generate massive combination of genetic diversity and may open the door to improved phenotypes based entirely on chromosome scale variations. In this work, we demonstrated that the ring_synV is an ideal genetic background to continuously generate phenotypic diversity by whole chromosome rearrangement. The PDV was selected as a marker to exhibit the phenotypic improvement, and we isolated candidates with ~7-fold increased and a near-wild-type growth.

The ring_synV tends to generate neochromosomes with complex SVs when exposed to estradiol. By SCRaMbLEing ring_synV, we isolate derivatives with a broad variety of phenotypes, facilitated by structural variant especially the large DNA segment duplication, and quite a high ratio of them are more productive than its initial strain. Though chromosome SVs were observed in the previous SCRaMbLE study, the scale and number are limited[22]. It seems that the topological structure of linear chromosome restricted the genetic rearrangement of DNA segments. In this work, we identified that the SCRaMbLE of ring_synV is capable to generate aneuploid chromosomes, which provides a platform for aneuploidy research. WGS revealed large-scale DNA duplications throughout the ring_synV SCRaMbLEants, indicating that the ring structure is a preferable genetic background to facilitate more genomic diversity, especially translocations, inversions and duplications, including inverted duplications, tandem duplication, translocated duplication, with head-to-head and tail-to-tail intrachromosomal fusions. The duplication, translocation, and inversion of large DNA segments occur in a semi-rational design method, which is based on site-specific recombination between loxPsym sites of SCRaMbLE system. In principle, the DNA segment could rearrange between every two loxPsym sites. However, the rearrangement events should be affected by some other factors, such as the chromosome topological structure. Though the classical genomic alteration approaches, such as random mutations, are able to generate strains with high productivity[29], the SCRaMbLE is a rapid method to continuously generate improved phenotypes and totally different chassis by chromosome rearrangement. Especially, the ring_synV chromosome is an ideal genetic background to show this performance.

Although the exact mechanism remains a matter of speculation, the ring chromosome generated much more complex genetic rearrangement events during SCRaMbLE. The change of chromosome topological organization in ring_synV should markedly decrease the steric hindrance between loxPsym sites, even located far away in linear distance. The simultaneous rearrangement between all of the 170 chromosomal segments may resulted in chromothripsis, which leads to CNVs as well as broad tandem duplications and translocated duplications intrachromosome[30]. Thus, much more breakage-fusion-bridge events can occur simultaneously at the designed loxPsym sites in ring_synV than the linear chromosome to form the dramatic SVs, termed chromothripsis, constitute an extreme complexity. Additionally, ring chromosome can have relatively high rates to form dicentric chromosomes, and the breaking of neochromosome during mitosis may lead to asymmetric segregation and changes in size[31]. Besides, a double rolling circle might be a mechanism to interpret the phenomenon of widespread duplications. Thus the complex genomic variation of ring_synV should emerge as a consequence of mechanisms including DNA recombination, replication and repair processes[32].

Genomic SV is implicated in phenotypic diversity, but it's elusive to dissect the mechanisms by which they exert their functional impact. SCRaMbLEing ring_synV facilitated more genomic diversity, especially duplication, translocation and inversion, and the isolates with topological changes have the capabilities for further SV. By analyzing the novel structures, we isolate colonies with increased PDV biosynthesis by selection in 11 of them though the exact mechanism is not clear. Meanwhile, the SVs can be continuously evolved, which is will of great benefit for the phenotypic improvement. Further analysis will be

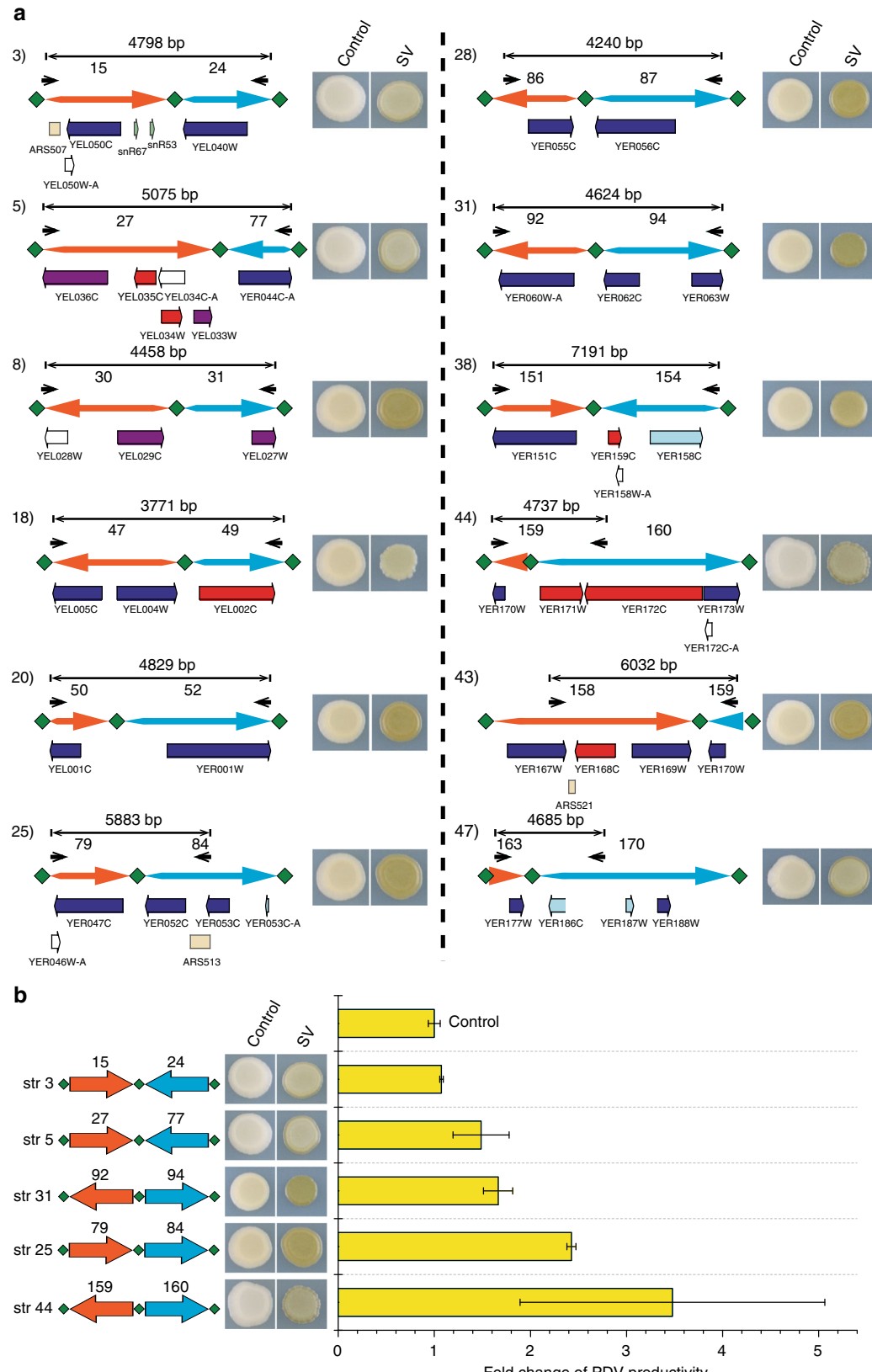

**Fig. 4** Functional novel SVs that may correlate with PDV production improvement. **a** Schematics showing novel SV that may correlate with PDV production improvement. The left DNA segment (orange arrow) and right DNA segment (blue) are divided by loxPsym sites (green diamond). The DNA sequence was amplified by using primers designed (black arrow) and transformed into yWJS1 and assembled with linearized pRS413 backbone plasmid. We isolated transformants with virtual darker colors by selection from 11 novel SVs than the initial strain. **b** Fold change of PDV production improvement related to novel SV structures that are quantitated by HPLC. Values are averages from three experiments, and error bars denote s.d

required to define these complex phenotypic relationships. In summary, the ring_synV exhibited the capability to continuously generate complex genotypes and phenotypes by SCRaMbLEing method. Our continuous approach by SCRaMbLEinging ring_synV embraces the evolution of genome by modifying the chromosome number, structure, and organization, and evolution of organisms with improved properties, identifying targets for phenotypic comprehension.

## Methods

**Strains and plasmids**. All *Saccharomyces cerevisiae* strains used in this study are listed in Supplementary Data 2. The strain yXZX923 carries a ring_synV chromosome[20].

The ORFs of *vioA*, *vioB*, and *vioE* were PCR amplified from BioBrick. All the transcription elements (promoters and terminators) were PCR amplified from BY4741 genomic DNA, which had 40-bp overlaps identical to the ends of *vioA*, *vioB*, *vioE*. The plasmid pWJ01 was assembled in yeast by co-transforming gel-purified *Eco*RI-digested pRS416 and the PCR amplicons followed by selection on synthetic complete medium lacking uracil (SC-Ura) (Supplementary Data 3). Phusion DNA polymerase and restriction enzymes were purchased from New England Biolabs. All oligonucleotides were purchased from Genewiz (Beijing, China).

**Integration of PDV pathway**. The chromosome evolution was applied by optimizing the production of PDV biosynthesis in a haploid yeast strain carrying a ring_synV. The integration of PDV pathway was performed by homologous recombination, directed by 500-bp *CAN1* genomic sequences flanking the *vioA-vioB-vioE-LEU2* construct[19].

The haploid yeast strain yXZX923, carrying a ring_synV, was transformed with the linearized plasmids. Before transformation, 1 μg of PDV pathway carrying plasmid (pWJ01) was digested with *Eco*RI in a final volume of 10 μL, as well as the left and right homologous arm carrying plasmids (pWJ02, pWJ03) were both digested with *Hind*III and *Sac*I. The whole digestion products were used directly for yeast transformation and LEU2$^+$ transformants were identified. The initial PDV expression strain yWJS1 was isolated with a PDV productivity of 0.23 mg/L.

**SCRaMbLE of ring_synV by GAL-Cre plasmid**. To avoid the leakiness of Cre-EBD activity in the absence of estradiol, a galactose inducible Cre expression plasmid (pCRE4) was utilized to induce the chromosome evolution[19]. The galactose and estradiol were used together to induce the expression of engineered Cre recombinase, which was fused to the murine oestrogen binding domain.

One single colony carrying the pCRE4 was picked from a SC-His plate and inoculated into 3 mL of SC-His medium at 30 °C for 24 h. Subsequently, cells were harvested and washed three times with sterile water, followed by inoculation in 3 mL SC medium without histidine and glucose, and containing 2% glycerol as carbon source medium overnight at 30 °C. The pellets were spun down, washed three times with sterile water, and resuspended in 3 mL SC medium without histidine and glucose, containing 2% galactose as carbon source medium and 1 mM of estradiol to an A600 = 0.6–1.0 for 24 h at 30 °C. Cells were taken before induction and at various time points after induction, and plated at SC-His plates with appropriate dilutions to make sure that similar numbers of colonies were observed on control and experimental (treated by galactose and estradiol) plates and incubated at 30 °C for 72 h, following with colony number counting and colony size measurements.

**Colorimetric and phenotypic screen**. Single colonies were picked into 3 mL of SC-His medium at 30 °C for 48 h, following with tenfold dilutions on SC-His plates, which were incubating at 30 °C for 72 h. Then scored for growth and photographed.

The cells of SCRaMbLEed populations were plated on SC-His plate at 30 °C and inoculated for 3 days to produce green colonies. Colonies with different color scores by visual inspection were counted. In total, six rounds of SCRaMbLE process were performed and six colonies with increased color intensity were screened for phenotypic and genetic analysis, representative.

**PCRTag analysis**. PCRTag analysis was a high-throughput method to analyze the presence, absence and CN of synthetic yeast chromosome segments (Supplementary Data 1)[20]. The SCRaMbLEd clones which have the optimized phenotypes (darker color) and a similar fitness with the initial strain (yWJS1) were picked for genomic DNA extraction and PCRTag analysis and qPCRTag analysis with a non-redundant array, using only one primer pair per loxPsym flanked segment. The absence of synthetic PCRTag amplicons (SYN) revealed the deletion of synV segments. The qPCRTag analysis results revealed the duplication of synV segments. TransFast Taq DNA Polymerase (TransGene Biotech), 200 nM each of forward and reverse primers, and gDNA of SCRaMbLEd ring_synV were used to amplify the PCR product in a 15 μL final volume. The following PCR program was used:

95 °C/5 min, 30 cycles (95 °C/30 s, 53 °C/30 s, 72 °C/30 s, 72 °C/10 min, 4 °C/∞. Detection of PCRTags was performed by agarose gel electrophoresis.

**HPLC analysis of PDV production**. HPLC analysis was applied for quantitatively measurement of PDV production. The selected colonies were incubated in SC medium at 30 °C overnight. Moderate overnight culture was added to fresh SC medium to OD$_{600}$ = 0.1 and incubated at 30 °C for 72 h. Each 2 mL of cultures was pelleted in a microcentrifuge for 6 min at 12,000 rpm. The pellets were resuspended in 1 mL of methanol and boiled at 75 °C for 25 min, overtaxing halfway through. After a 6 min of centrifugation at 12,000 rpm, the supernatant was analyzed by HPLC[33].

**Whole-genome sequencing**. WGS was carried out by Frasergen (Wuhan, China). Paired-end WGS of strains yWJS1, yWJS044, yWJS047, yWJS067, yWJS184, and yWJS321 were performed using an Illumina HiSeq. Raw reads were obtained and used for downstream analysis. Briefly, reads were first mapped using BWA with default parameters; a reference genome was constructed with the sequence for strain. Duplication was visualized in IGV browser. Base changes and short indels were detected using the HaplotypeCaller function of the Genome Analysis Toolkit (GATK) with standard parameters. SV was detected using Break Dancer. Native sequence was detected by realigning the reads rejected by BWA in the first round against a reference genome containing the native chromosome(s) of interest. Read counts for each chromosome were determined from WGS bam files using Bedtools "genome coverage." Chromosome CN was then calculated by generating boxplots in R using ggplot2. Paired-end reads were mapped to ring_synV chromosome. Then reads containing loxPsym site that were not mapped to ring_synV were picked out and trisected at loxPsym site into two short ends, which remain associated with the paired sequence from the other end of the fragment. Two paired short ends were aligned to ring_synV using bowtie2 single-end mapping to discover new loxP-sym junctions and structure after SCRaMbLE, with a depth of at least three reads.

For PacBio sequencing, isolation of genomic DNA was carried out using SDS method. Total DNA obtained was subjected to quality control by agarose gel electrophoresis and quantified by Qubit. Standard 10 kb-insert protocols were used for PacBio sequencing and barcode adapter was used to construct SMRT bell sequencing library. Sequencing was carried out on the Pacific Bioscience Sequel platform using Sequel Binding Kit 2.0, Sequel Sequencing Kit 2.1 and Sequel SMRT Cell 1 M v2 reagents with 600 min movies.

**Stability analysis**. A series of SCRaMbLEants were selected to test the stability of phenotypic (Supplementary Fig. 5). One single colony of diploid was picked up from SC-Leu plates and inoculated in 5 mL of SC-His medium at 30 °C overnight and then 5 mL of overnight culture was transferred to 5 mL of fresh YPD medium. After ~60 generations inoculation in SC-His medium, 10-fold serial dilution assays were used to analyze the stability of ring chromosome.

**Statistical analysis**. No statistical methods were used to predetermine sample sizes. The SCRaMbLEd yeast colonies were picked by different colony colors which were pre-established. Data collection and analysis were not randomized nor performed blind to the conditions of the experiments. No data points were excluded. The data are presented as mean ± s.d. and statistical differences were determined using unpaired *t*-test. Statistical significance was set as *$P < 0.05$, **$P < 0.005$, and ***$P < 0.001$. The excel was used for the statistical analyses.

## Data availability

The data that support the findings of this study are available from the corresponding author on request. All genomic data for this paper have been deposited in GeneBank (https://www.ncbi.nlm.nih.gov/genbank/) that are available from the Sequence Read Archive under Accession Code SRP158116 as well as the BIG (http://bigd.big.ac.cn/) ?under the BioProject Accession No. CRA000963.

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

## Acknowledgements

This work was funded by the National Natural Science Foundation of China (21621004, 21750001) and the Ministry of Science and Technology of China ("973" Program, 2014CB745100). We thank Bo-Han Chen for the assistant analysis of genome sequencing data.

## Author contributions

J.W., Z.-X.X., B.J., and Y.-J.Y. conceived the study and designed all experiments. J.W. worked on the SCRaMbLE experiments. Z.-X.X. and Y.M. performed sequencing analysis. Z.-X.X., X.-R.C., Y.-Q.H., B.H., and Y.M. focused on the phenotypic analysis of novel SVs. J.W., Z.-X.X., and B.-Z.L. oversaw all data analysis. Z.-X.X. and Y.-J.Y. drafted the manuscript with topical input from all other authors.

## Additional information

**Competing interests:** The authors declare no competing interests.

