## [Peer Review File · Nature Communications]

Reviewers' Comments:

Reviewer #1:

Remarks to the Author:

This manuscript described the implementation of a previously published method named SCRaMbLE that relies on Cre-loxP to introduce large-scale genome deletions, inversions, translocations, and duplications, etc. They chose the previously synthesized chromosome 5, which is in the BAC-based circular form and contains 170 loxP sites following nonessential genes. Taking prodeoxyviolacein (PDV) biosynthesis as an example, they showed that complex genome variations could improve the production of PDV significantly. Genome sequencing revealed that the generation of some novel structural variations is related to the promotion of the PDV biosynthesis.

This work contains some interesting results. However, the current version needs to be carefully revised due to the obscurity in many places. Some details might have been mentioned in the series of papers related to SCRaMbLE technology, but the readability of this single article needs to be improved.

1. In Line 47-L53, the authors mentioned that SCRaMbLing the ring_synV could generate many more structural rearrangements than if it is done on the linear synV. A couple of sentences are needed here to support this claim. The authors had some illustrations in the Discussion. Adding a brief explanation in the Introduction will be very helpful.
2. In line 56, telomere position effect was eliminated in ring_synV. Why is it important? Need 1-2 short sentences here for clarity. Is this elimination the only difference between ring_synV and synV?
3. The sentences starting from Line 91 Figure S1: (1) When illustrating a figure with several subfigures, each of the subfigures needs to be marked in the text when it is mentioned. This is a common issue repeatedly occurring across the entire manuscript. (2) The subfigures need to be presented in the same order as they are introduced in the text. There are many situations that the subfigure c (e.g., Figure S2c) was presented before the discussion on subfigure a (e.g., Figure S2A). Check the order of Figure S4A and S4B as well.
4. Line 94: the authors mentioned 'we observed 3.5%-40.9% colonies were color burn'. What does 'color burn' mean? How was this range of 3.5%-40.9% calculated? It seemed Figure S1C was mislabeled. It should be for SynV instead of ring_synV.

In the caption of Figure S1a, for the same time point, were the patches of colonies from left to right correlated with different dilutions?
5. Throughout the manuscript, no statistical analysis was performed, which made some conclusions on the percentage changes lack STATS base.
6. Line 134: It was mentioned 'Notably, color deepened colonies from yWJS047 was recovered about 2 to 14 times more efficiently' when the second round SCRaMbLE was performed.' It was not clear how this 2-14 times was obtained. Secondly, why strain 607 was picked instead of 047?
7. Line 166: explain 'unbalanced form variation'.
8. Line 176: what does 'internal sequences' mean in the context of 'the genomic rearrangement do not affect internal sequences to the loxPsym sites'?
9. Quite a few figures were missing, including Figure 3C (Line 183), Figure 3D (Line 188), Figure 3E (Line 199).

10. Line 187: the authors particularly mentioned 'applied PacBio sequencing analysis for yWJS067'. What other sequencing method was used for other strains?

11. Line 204: explain 'blank DNA segments'. The authored mentioned, 'of the 170 segments divided by loxPsym sites, 31 contain essential genes, 14 are blank DNA segments, and 22 contains no PCRTags'. What about the rest 103 (170-31-14-22)?

12. Overall speaking, more information is needed in the legends of figures, especially when the data was not presented in the common formats (e.g., Figure 2c, 3a). What does the coverage depth mean?

Also, there are many strains in Figure 2A, but it is not clear where they were from.

Other minor changes are listed below:

1. Line 32, change 'implicated' to 'are implicated'.
2. Line 72-73 is just a repeat of the previous sentence. Please twist the writing.
3. What is the function of estradiol in this manuscript?
4. Line 92: what is the relative strain?
5. Line 112: change 'yield' to 'yielded'.
6. Line 116: change 'may be not' to 'were not'.
7. Line 137: change 'to deep reveal the genetic variations' to 'to deeply reveal'.
8. When 'aneuploidy' was first mentioned, define it.
9. Line 153: change 'predict the accurately copy number variation' to 'accurately predict the copy number'.
10. Line 162: CNVs need to be defined at the first sight.
11. Line 194: consider changing 'make sure' to 'explore'.
12. Line 209: 'although in several cases annotated phenotypes of gene deleted did not appear sufficient to cause the observed phenotypes'. Should it be 'although in several cases annotated GENOTYPES of gene DELETION...'?

Reviewer #2:

Remarks to the Author:

Authors applied SCRaMbLE method to induce mutations for a better PDV mutant. It looks like another mutation method. SCRaMbLE seems not to be a new method, it has been used to construct artificial yeast chromosome. Following concerns need be addressed:

Issues of concerns:

1. The title contains too many abbreviations, it is confusing;
2. Please describe briefly inducible SCRaMbLE system compared with genome shuffling and random mutations, their advantages and disadvantages;
3. The large DNA segment duplication, translocation and inversion occur randomly? or selectively? Could this be random mutations? Please discuss;
4. Page 4, lines 94-95: "Nearly 3.67% of ring_synV and 2.61% of synV survivals seems to have an optimized PDV production,...". Questions: from how many survival colonies?
5. Pages 4-5, lines 94-109: which figures these statements refer to?
6. Page 5, lines 110-124 (second paragraph): "After six cycles of SCRaMbLE exhibition, we got an

isolate with more than sevenfold increase in PDV production as well as a near-wild-type growth (Figure 2A).": This is very impressive. But what was the start PDV concentration? Could random mutation achieves the same? Over even more effectively?

7. SCRaMbLE seems to be effective. However, PDV is not a typical product for yeast. Could SCRaMbLE be applied for increasing ethanol production by yeast?

8. Page 7, lines 187-188: "The result revealed complicated inverted duplications, tandem duplication and translocated duplications (Figure 3D)." Why there were not homologous recombinations happen?

9. Authors should take the best PDV producer and DNA sequence it, to see how the genome has been changed by SCRaMbLE.

Dear Editor,

We would like to thank you and referees for giving us constructive suggestions. Here we submit the manuscript, modified according to your suggestions. Our responses to the reviewers' comments were outlined as follows:

Referee #1:

This manuscript described the implementation of a previously published method named SCRaMbLE that relies on Cre-loxP to introduce large-scale genome deletions, inversions, translocations, and duplications, etc. They chose the previously synthesized chromosome 5, which is in the BAC-based circular form and contains 170 loxP sites following nonessential genes. Taking prodeoxyviolacein (PDV) biosynthesis as an example, they showed that complex genome variations could improve the production of PDV significantly. Genome sequencing revealed that the generation of some novel structural variations is related to the promotion of the PDV biosynthesis.

This work contains some interesting results. However, the current version needs to be carefully revised due to the obscurity in many places. Some details might have been mentioned in the series of papers related to SCRaMbLE technology, but the readability of this single article needs to be improved.

Comment 1:

In Line 47-L53, the authors mentioned that SCRaMbLing the ring_synV could generate many more structural rearrangements than if it is done on the linear synV. A couple of sentences are needed here to support this claim. The authors had some illustrations in the Discussion. Adding a brief explanation in the Introduction will be very helpful.

Response:

We added a brief explanation in the Introduction. Please see Line 49-51.

Comment 2:

In line 56, telomere position effect was eliminated in ring_synV. Why is it important? Need 1-2 short sentences here for clarity. Is this elimination the only difference between ring_synV and synV?

Response:

The elimination of telomeres are the only difference between the ring_synV and synV, we added sentences for clarity in the manuscript, please see Line 59. The elimination of TPE could change gene expression and it might be an incentive to raise the rearrangement of ring chromosome together with the chromosome organization rebuild. It worth a further research later.

Comment 3:

The sentences starting from Line 91 Figure S1: (1) When illustrating a figure with several subfigures, each of the subfigures needs to be marked in the text when it is mentioned. This is a common issue repeatedly occurring across the entire manuscript. (2) The subfigures need to be presented in the same order as they are introduced in the text. There are many situations that the subfigure c (e.g., Figure S2c) was presented before the discussion on subfigure a (e.g., Figure S2A). Check the order of Figure S4A and S4B as well.

Response:

Thanks for your suggestion, we changed all the subfigures introduced in the text.

Comment 4:

Line 94: the authors mentioned 'we observed 3.5%-40.9% colonies were color burn'. What does 'color burn' mean? How was this range of 3.5%-40.9% calculated? It seemed Figure S1C was mislabeled. It should be for SynV instead of ring_synV.

In the caption of Figure S1a, for the same time point, were the patches of colonies from left to right correlated with different dilutions?

Response:

We changed “color burn” to “darker color” in Line 119.

We added description in the manuscript for the calculation, please see Line 97-98 and 117-119.

We changed the figure caption to “synV” in of Supplementary Fig. 1c.

In Supplementary Fig. 1a, the patches of colonies from left to right were all serially diluted in 10-fold increments in water and plated onto SC-His media and incubated at 30°C for 2 days. We added the description in the figure caption.

Comment 5:

Throughout the manuscript, no statistical analysis was performed, which made some conclusions on the percentage changes lack STATS base.

Response:

We have done the statistical analysis, please see Supplementary Fig. 2a and 2c.

Comment 6:

Line 134: It was mentioned ‘Notably, color deepened colonies from yWJS047 was recovered about 2 to 14 times more efficiently’ when the second round SCRaMbLE was performed.’ It was not clear how this 2-14 times was obtained. Secondly, why strain 067 was picked instead of 047?

Response:

We described the 2 to 14 times in the manuscript, please see Line 123-125.

We described the reason for picking strain yWJS067 in the manuscript, please see Line 127-130.

Comment 7:

Line 166: explain ‘unbalanced form variation’.

Response:

“Unbalanced form variation” means the structural variations which lead to copy number variation, such as duplication and deletion. We added a description in Line 22.

Comment 8:

Line 176: what does ‘internal sequences’ mean in the context of ‘the genomic rearrangement do not affect internal sequences to the loxPsym sites’?

Response:

It means the DNA sequence between two loxPsym sites. We described it in manuscript, please see Line 189.

Comment 9:

Quite a few figures were missing, including Figure 3C (Line 183), Figure 3D (Line 188), Figure 3E (Line 199).

Response:

All the figure numbers were corrected.

Comment 10:

Line 187: the authors particularly mentioned ‘applied PacBio sequencing analysis for yWJS067’. What other sequencing method was used for other strains?

Response:

We applied whole-genome sequencing analysis for all the strains mentioned in Fig. 2c, and applied PacBio sequencing to analyze the complex structural variations in the SCRaMbLED strain yWJS067.

Comment 11:

Line 204: explain 'blank DNA segments'. The authored mentioned, 'of the 170 segments divided by loxPsym sites, 31 contain essential genes, 14 are blank DNA segments, and 22 contains no PCRTags'. What about the rest 103 (170-31-14-22)?

Response:

We described the "blank DNA segments" in the text, please see Line 217.

We described the "rest 103 segments" in the manuscript, please see Line 218-219 and Supplementary Table 1.

Comment 12:

Overall speaking, more information is needed in the legends of figures, especially when the data was not presented in the common formats (e.g., Figure 2c, 3a). What does the coverage depth mean?

Response:

Thanks for your comments and we added details in the figure captions.

The coverage depth stands for the chromosome copy number which were determined from WGS result by using Bedtools "genome coverage". Please see the figure caption of Fig. 2c and Line 365-367 in the text.

Comment 13:

Also, there are many strains in Figure 2A, but it is not clear where they were from.

Response:

We listed all of the strains in Supplementary Table 2.

Other minor changes are listed below:

Comment 14:

Line 32, change 'implicated' to 'are implicated'.

Response:

It was changed.

Comment 15:

Line 72-73 is just a repeat of the previous sentence. Please twist the writing.

Response:

We twisted the writing, please see Line 75-77 in the manuscript.

Comment 16:

What is the function of estradiol in this manuscript?

Response:

The estradiol was used to induce the expression of engineered Cre recombinase, which was fused to the murine oestrogen binding domain. We added an explanation in Line 323-324.

Comment 17:

Line 92: what is the relative strain?

Response:

We changed the “relative strain” to “parent strain”, please see Line 96.

Comment 18:

Line 112: change ‘yield’ to ‘yielded’.

Response:

It was changed.

Comment 19:

Line 116: change ‘may be not’ to ‘were not’.

Response:

It was changed.

Comment 20:

Line 137: change ‘to deep reveal the genetic variations’ to ‘to deeply reveal’.

Response:

It was changed.

Comment 21:

When ‘aneuploidy’ was first mentioned, define it.

Response:

We added a definition, please see Line 154.

Comment 22:

Line 153: change ‘predict the accurately copy number variation’ to ‘accurately predict the copy number’.

Response:

It was changed.

Comment 23:

Line 162: CNVs need to be defined at the first sight.

Response:

It was defined at the first sight, please see Line 173.

Comment 24:

Line 194: consider changing ‘make sure’ to ‘explore’.

Response:

Thank you for the comment, we changed the “make sure” to “explore”.

Comment 25:

Line 209: ‘although in several cases annotated phenotypes of gene deleted did not appear sufficient to cause the observed phenotypes’. Should it be ‘although in several cases annotated GENOTYPES of gene DELETION...’?

Response:

We rewrite it as your suggestion, please see Line 223-224.

Reviewer #2 (Remarks to the Author):

Authors applied SCRaMBLE method to induce mutations for a better PDV mutant. It looks like another mutation method. SCRaMBLE seems not to be a new method; it has been used to construct artificial yeast chromosome. Following concerns need be addressed:

Issues of concerns:

Comment 1:

The title contains too many abbreviations; it is confusing;

Response:

Thanks for your comment and we changed it.

Comment 2:

Please describe briefly inducible SCRaMbLE system compared with genome shuffling and random mutations, their advantages and disadvantages;

Response:

We compared the SCRaMbLE system with genome shuffling and random mutations in Discussion. Please see Line 233-243.

Comment 3:

The large DNA segment duplication, translocation and inversion occur randomly? or selectively? Could this be random mutations? Please discuss;

Response:

The SCRaMbLE is a semi-rational designed system. We added a description in the Discussion, please see Line 262-266.

Comment 4:

Page 4, lines 94-95: "Nearly 3.67% of ring_synV and 2.61% of synV survivals seems to have an optimized PDV production,...". Questions: from how many survival colonies?

Response:

We added this data, please see Supplementary Fig. 2a, 2c, and Supplementary Table 2.

Comment 5:

Pages 4-5, lines 94-109: which figures these statements refer to?

Response:

These statements refer to Supplementary Fig. 1, we added it.

Comment 6:

Page 5, lines 110-124 (second paragraph): "After six cycles of SCRaMbLE exhibition, we got an isolate with more than sevenfold increase in PDV production as well as a near-wild-type growth (Figure 2A)": This is very impressive. But what was the start PDV concentration? Could random mutation achieves the same? Over even more effectively?

Response:

The start PDV concentration in initial strain yWJS1 is 0.23 mg/L. We added it into Line 319.

We think random mutation is able to improve the PDV production according to the previous research, though we are not sure about the capability. Compared to the random mutations, the SCRaMbLE is a rapid method to continuously generate improved phenotypes and totally different chassis by chromosome rearrangement. Especially, the ring_synV chromosome is an ideal genetic background to show this performance. We added a description in the manuscript, please see Line 267-270.

Comment 7:

SCRaMbLE seems to be effective. However, PDV is not a typical product for yeast. Could SCRaMbLE be applied for increasing ethanol production by yeast?

Response:

The PDV is not only a precursor of potential anti-cancer drug violacein, but also a downstream metabolite of aromatic amino acids (AAA) which are important precursors to many high value-added biochemicals, drugs and nutraceuticals. We added the description in Line 62-66 of the manuscript.

We have not applied the SCRaMbLE strategy to increase ethanol production in yeast yet. But the publish paper by Luo et al (Identifying and characterizing SCRaMbLED synthetic yeast using ReSCuES, Nat Commun, 2018, 9(1); DOI: 10.1038/s41467-017-00806-y) demonstrated that the

SCRaMbLE is capable to generate ethanol tolerant yeast strain, thus we think SCRaMbLE could be applied for increasing ethanol production by yeast.

Comment 8:

Page 7, lines 187-188: "The result revealed complicated inverted duplications, tandem duplication and translocated duplications (Figure 3D)." Why there were not homologous recombinations happen?

Response:

We added the explanation in the manuscript, please see Line 43-46 and 262-265.

Comment 9:

Authors should take the best PDV producer and DNA sequence it, to see how the genome has been changed by SCRaMbLE.

Response:

The best PDV producer (yWJS321) was analyzed by whole-genome sequencing to detect the structural variations (Fig. 2c and 2d). The aneuploid chromosome variation was represented in Fig. 2c, the copy number variation was represented in Fig. 2d, and the novel structural variations were represented in Fig. S3.

Thank you very much for your kindness and help again! We would be glad to provide any additional information if needed, please feel free to contact us.

Yours sincerely,

Ying-Jin Yuan

Professor of the biochemical engineering

Fellow of IChemE

Director, Key Laboratory of Systems Bioengineering, Ministry of Education

PI, SynBio Research Platform, Collaborative Innovation Center of Chemical Science and Engineering (Tianjin)

School of chemical engineering and Technology,

Tianjin University, Tianjin 300072

P.R. China

Tel: 86-22-27401288

Fax:86-22-27403389

E-mail: yjyuan@tju.edu.cn

Reviewers' Comments:

Reviewer #1:

Remarks to the Author:

The authors have made sufficient modifications to the manuscript to improve the clarity of the writing. The revised version has the quality for publication.

Reviewer #2:

Remarks to the Author:

Authors have addressed my concerns well, with new data. I suggest to accept the paper in its current form.

REVIEWERS' COMMENTS:

Reviewer #1 (Remarks to the Author):

The authors have made sufficient modifications to the manuscript to improve the clarity of the writing. The revised version has the quality for publication.

Reviewer #2 (Remarks to the Author):

Authors have addressed my concerns well, with new data. I suggest to accept the paper in its current form.

Response:

We deeply appreciate the reviewers' editing and comments to improve this manuscript.